# Validity of the Japanese Version of the Quick Mild Cognitive Impairment Screen

**DOI:** 10.3390/ijerph16060917

**Published:** 2019-03-14

**Authors:** Ayako Morita, Rónán O’Caoimh, Hiroshi Murayama, D. William Molloy, Shigeru Inoue, Yugo Shobugawa, Takeo Fujiwara

**Affiliations:** 1Department of Global Health Promotion, Tokyo Medical and Dental University, Tokyo 113-8510, Japan; fujiwara.hlth@tmd.ac.jp; 2Clinical Sciences Institute, National University of Ireland Galway, H91 TK33 Galway City, Ireland; ronan.ocaoimh@nuigalway.ie (R.O.); W.Molloy@ucc.ie (D.W.M.); 3Centre for Gerontology and Rehabilitation, University College Cork, St Finbarrs Hospital, Douglas road, T12 XH60 Cork City, Ireland; 4Institute of Gerontology, The University of Tokyo, Tokyo 113-8656, Japan; murayama@iog.u-tokyo.ac.jp; 5Department of Preventive Medicine and Public Health, Tokyo Medical University, Tokyo 160-8402, Japan; inoue@tokyo-med.ac.jp; 6Division of International Health, Graduate School of Medical and Dental Sciences, Niigata University, Niigata 951-8122, Japan; yugo@med.niigata-u.ac.jp

**Keywords:** cognitive impairment, short cognitive screen instrument, Japanese, older adults, screening

## Abstract

Early detection of dementia provides opportunities for interventions that could delay or prevent its progression. We developed the Japanese version of the Quick Mild Cognitive Impairment (Q*mci*-J) screen, which is a performance-based, easy-to-use, valid and reliable short cognitive screening instrument, and then we examined its validity. Community-dwelling adults aged 65–84 in Niigata prefecture, Japan, were concurrently administered the Q*mci*-J and the Japanese version of the standardized Mini-Mental State Examination (sMMSE-J). Mild cognitive impairment (MCI) and dementia were categorized using established and age-adjusted sMMSE-J cut-offs. The sample (n = 526) included 52 (9.9%) participants with suspected dementia, 123 (23.4%) with suspected MCI and 351 with likely normal cognition. The Q*mci*-J showed moderate positive correlation with the sMMSE-J (*r* = 0.49, *p* < 0.001) and moderate discrimination for predicting suspected cognitive impairment (MCI/dementia) based on sMMSE-J cut-offs, area under curve: 0.74, (95%CI: 0.70–0.79), improving to 0.76 (95%CI: 0.72 to 0.81) after adjusting for age. At a cut-off of 60/61/100, the Q*mci*-J had a 73% sensitivity, 68% specificity, 53% positive predictive value, and 83% negative predictive value for cognitive impairment. Normative data are presented, excluding those with any sMMSE-J < 27. Though further research is required, the Q*mci*-J screen may be a useful screening tool to identify older adults at risk of cognitive impairment.

## 1. Introduction

Worldwide, 46.8 million people are living with dementia, and this number is increasing [1]. Associated global medical and social care costs are estimated to be US $818 billion annually, and are projected to double by 2035 [1,2]. Epidemiological research suggests that potentially modifiable cardiovascular and lifestyle factors contribute to dementia risk [3], and multi-dimensional interventions are effective in delaying the progression of cognitive decline [4,5].

Timely and accurate diagnosis of impairment at its earliest stage is important in order to benefit from these interventions, and subtle cognitive impairment, typically proceeding several years before overt clinical symptoms, may be a useful indicator to target preventative strategies [6].

A screening test aims at determining the likelihood of having the disease among people who as yet have no symptoms. The most commonly used brief cognitive screening instrument (CSI) is the Mini-Mental State Examination (MMSE) and the standardized MMSE (sMMSE), which have been translated into various languages, including Japanese [7,8]. The sMMSE is a 30-point test that assesses a board range of major cognitive domains (i.e., orientation to time and place, registration, attention and calculation, recall, language, repetition, and complex commands). However, despite high specificity (99.0%), it has low sensitivity (41.0%) in discriminating older adults with normal cognition and mild cognitive impairment (MCI) when carried out in a Japanese memory clinic using a cut-off of ≤26 [8,9]. In addition, this examination is considered overly long and complex for routine use in primary care [10]. While most clinical examinations in primary care last less than 10 min [11], the sMMSE typically takes longer, up to 15 min in those with dementia [12], such that briefer and simpler tests are recommended for use in busy community settings [10,13,14].

Recently, the Quick Mild Cognitive Impairment (Q*mci*) screen has been validated as an easy-to-use, rapid (<5 min), and reliable CSI for early detection of cognitive impairment [15,16]. The Q*mci* screen is highly sensitive and specific in differentiating MCI from normal cognition and dementia [15]. It was designed to minimize the floor and ceiling effects associated with the MMSE, as well as more MCI-specific CSIs such as the Montreal Cognitive Assessment (MoCA) [15,16]. Since its development and validation in Canada [16], it has been translated and validated in various languages worldwide, showing high sensitivity (0.76–0.92) and specificity (0.72–0.95) for differentiating MCI from normal cognition in a range of settings [17,18,19,20]. A recent systematic review and meta-analysis found that it compares favorably with other short CSIs specific for MCI [21]. Its primary advantage over many other similarly accurate instruments is its brevity [15]; the use of a brief but accurate CSI could help improve the efficient use of limited acute hospital services by triaging those most appropriate for further follow-up.

The present study aims to translate, validate (concurrent with the Mini-Mental State Examination—Japanese version—sMMSE-J) and provide normative data for the Japanese version of Q*mci* screening among older community-dwelling Japanese adults.

## 2. Materials and Methods 

### 2.1. Translation of the *Q*mci Screen

The original English version of the Q*mci* screen was translated into Japanese by a bilingual health researcher, fluent in English, but a native Japanese speaker. The translation was reviewed by an expert panel including the original translator, Japanese health professionals specializing in older people, and researchers with extensive experience in cognitive test development and translation. The complete version was then back-translated into English by a non-expert panel member, a native English speaker proficient in Japanese. The final version of the test was then piloted with five older Japanese community-dwellers and reviewed and approved by the expert panel for conceptual, cultural and linguistic equivalence.

Like the original English version, the Q*mci*-J screen comprises of six subtests assessing six domains of cognition: (1) Orientation, (2) word registration (five word), (3) clock drawing, (4) delayed recall, (5) verbal fluency (category) and (6) logical memory (immediate verbal recall of a short story); the total score ranges between 0 (indicating marked impairment) and 100 (suggesting no impairment). Cut-off scores adjusted for age and education are published for those presenting with cognitive symptoms (23).

Some changes were required, for example in word registration and recall we replaced “love (*a-i*)” with “sense of security (*an-sin*)” to match the linguistic frequency of the other four words (i.e., dog, rain, butter, door) in a Japanese cultural context, considering the number of syllables and category of the original. We also replaced “fragrant (*ka-o-ri-no-yo-i*)” with “beautiful (*ki-re-i-na*)” in the logical memory test for the same reason.

### 2.2. Study Settings and Participants

This study was conducted as a part of the Neuron to Environmental Impact across Generation (NEIGE) study, a cohort study that aims to explore determinants of healthy ageing and longevity in rural areas of Japan. Between September and October 2017, 600 community-dwelling adults aged between 65 and 85 years old, randomly sampled in Tokamachi city (Niigata prefecture, Japan), were invited to participate in a face-to-face survey and health check-ups at community centers. Of these, 526 (87.7%) agreed to participate, and completed the Q*mci*-J screen and sMMSE-J as a part of the broader study. Both instruments were administered consecutively, but in random order.

### 2.3. Data Analysis

Descriptive statistics were used to summarize data. In the first analysis (scenario 1) cognition was classified using the established cut-offs for MCI and dementia on the sMMSE-J: According to scoring ranges provided in the sMMSE-J manual, participants were classified into likely normal cognition (sMMSE-J score: 27–30), suspected MCI (sMMSE-J score: 21–26) and suspected dementia (sMMSE-J score: ≤20). In the second analysis (scenario 2), cognition was classified according to age cut-offs for mean sMMSE-J scores, which were applied to the data. Education cut-offs were not used as the sMMSE-J is not overly impacted by this [22]. Taking 95% of the population’s results as a normal reference range [23], we defined and removed values as outliers if they fell beyond the mean minus two standard deviations (SD) of sMMSE-J scores in those with normal cognition and MCI (the 2.5% of values at the bottom end of the normal range), and created new age-adjusted cut-offs for those aged 65–74 years (range for likely normal: 26.6–30, suspected MCI: 20.3–26.6, and suspected dementia: <20.3) and for subjects aged ≥75 (normal: 25.8–30, MCI: equal to or greater than 21.2, <25.8, dementia: less than 21.2). In the third analysis (scenario 3), normative data for the Q*mci*-J screen were examined for: those only scoring in the established normal range according to the sMMSE-J manual cut-off (i.e., ≥27) (scenario 3a), and those scoring ≥27 on the sMMSE-J, excluding those with subjective (self and informant reported) memory complaints, impairment in activities of daily living (ADL), history of diagnosed dementia, Parkinson’s disease, stroke, those currently receiving treatment of depression or other psychiatric medication (scenario 3b). Participant characteristics were examined using the Chi square-test for categorical variables and the two-way analysis of variance (ANOVA) test for continuous variables. Spearman’s correlation coefficients were calculated to examine correlations between the Q*mci*-J screen and sMMSE-J scores (both were positively skewed), and evaluate the concurrent validity of the Q*mci*-J. For an evaluation of diagnostic agreement with the sMMSE-J, we calculated the accuracy of the Q*mci*-J screen for predicting subjects with suspected cognitive impairment (suspected MCI and dementia) from the area under the curve (AUC) of receiver-operator curves (ROC) with 95% confidence intervals (CI). Finally, the maxim agreement with the MMSE-J classification was identified by computing the sensitivity, specificity, positive predictive value (PPV) and negative predictive value (NPV) at different Q*mci*-J cut-offs. The optimal cut-off was selected using Youden’s Index (J = Sensitivity + Specificity − 1). All statistical analyses were performed using STATA 14.0 (StataCorp LLC, College Station, TX, USA).

### 2.4. Ethical Considerations

All subjects provided written, informed consent for inclusion before they participated in the study. The study was conducted in accordance with the Declaration of Helsinki (1964–2013), and the protocol was reviewed and approved by the Institutional Review Board of Niigata University (2666) and Tokyo Medical and Dental University (M2017-233).

## 3. Results

The present study included 526 participants with a mean age of 73.5 and SD of ±5.6 years. Most, 52.7%, were female. The largest proportion had at least completed secondary school education, 44.7%. Less than 5% had hearing or visual impairment. Only 3 participants (0.57%) had an established diagnosis of dementia. The median score for the Q*mci*-J screen was 62, with an interquartile range (IQR) of 55 to 68. Its median administration time was 6 min. The characteristics of all participants included, and according to their categorization in scenario 1, are presented in Table 1.

### 3.1. Scenario 1

Using established cut-offs for the sMMSE-J, 351 participants were classified with likely normal cognition (median sMMSE-J score of 28/30, IQR 27–29), 159 were suspected MCI cases (median 25, IQR 24–26), and 16 others were suspected dementia cases (median 19, IQR 17–20). Participants who were older (*p* < 0.001), less educated (*p* < 0.001) and had impaired hearing (*p* < 0.001) were more likely to be classified as suspected cognitive impairment (MCI or dementia) but sex (*p* = 0.051) and visual impairment (*p* = 0.89) were not associated with the classification (Table 1). The Q*mci*-J screen scores were moderately and positively correlated with the sMMSE-J, where positive scores indicate higher cognitive function in both tests (Spearman’s *r* = 0.49, *p* < 0.001) (Figure 1). 

The accuracy of the Q*mci*-J screen was moderate to good with an area under the curve (AUC) of 0.74 (95%CI: 0.70–0.79) for predicting suspected MCI and dementia, identified using established sMMSE-J cut-off scores, and was excellent for predicting suspected dementia, AUC of 0.90 (95%CI: 0.84–0.97) (Figure 2). Table 2 and Table 3 show the sensitivity, specificity, PPV, NPV of the Q*mci*-J screen with 95% CIs for predicting suspected cognitive impairment (MCI and dementia) identified by the sMMSE-J at different Q*mci*-J screen cut-offs. Using Youden’s Index, 60/61 was found to be the optimal the cut-off point for differentiating suspected cognitive impairment (MCI + dementia based on sMMSE-J scores), producing a sensitivity of 73%, specificity of 68%, PPV of 53% and NPV of 83% (Table 2). A cut-off point of 56/57 was best for discriminating between suspected dementia and non-dementia (normal and those with suspected MCI), with a sensitivity of 94.0%, specificity of 72.0%, PPV of 10.0% and NPV of 100.0% (Table 3).

### 3.2. Scenario 2 (Using sMMSE-J Age-Adjusted Cut-Offs)

When sMMSE-J scores were adjusted for age, 260 participants were classified as having likely normal cognition, 222 with suspected MCI and 30 with suspected dementia. Using this classification, the accuracy of the Q*mci*-J screen for separating likely normal cognition from suspected cognitive impairment (MCI or dementia) improved to an AUC of 0.76, 95% CI 0.72 to 0.81. Similarly, its accuracy for differentiating suspected dementia from non-dementia improved to an AUC of 0.91, 95% CI 0.86 to 0.97.

### 3.3. Scenario 3a (Normative Data Based on sMMSE-J Scores ≥27 Only)

Normative values for the Q*mci*-J screen were developed based on scores grouped by age and education for participants with sMMSE-J scores ≥27 only (n = 352) included. These are presented in the Table 4 and show how Q*mci*-J screen scores vary by an individual’s age and education, increasing in those with more education, and decreasing in those with greater age, except for those aged 80–84 years who had the lowest scores, but were not affected by their number of years in education.

### 3.4. Scenario 3b (Normative Data Based on sMMSE-J Scores ≥27 in Those Without Cognitive Symptoms or Supportive Features)

Finally, we present the normative values of the Q*mci*-J screen by age and education for participants with sMMSE-J scores ≥27, but without a history of dementia, subjective memory complaints, ADL impairment or other features or conditions associated with cognitive decline (n = 265). These were broadly similar to those developed with scores ≥27, and are presented in the Table 5.

## 4. Discussion

The present study is the first to develop and evaluate a Japanese version of the Q*mci*-J screen to aid in the screening of community-dwelling older adults, and determining their likelihood of having cognitive impairment. It presents the concurrent validity of the Q*mci*-J screen against the established sMMSE-J using different modeling exercises (scenarios), and shows that the Q*mci*-J screen score was moderately, positively and significantly correlated with the sMMSE-J. A Q*mci*-J screen cut-off of 60/61 out of 100 points provided a 73% sensitivity and 68% specificity for differentiating older adults with suspected cognitive impairment (MCI and dementia) and likely normal cognition according to established cut-offs for these on the sMMSE-J.

Time constraints are a common and important barrier to using CSIs in busy primary care settings [21,24], though it is recognized that they must be interpreted with care, and are not a substitute for clinical assessment [24]. The median time to administer the Q*mci*-J screen was 6 min, longer than the previous reports of 3–5 min [22,25], but still acceptable. This additional time may in part be due to our study procedure, where we asked participants to name their highest educational institution to verify if it was an educational institution accredited by the Japan Ministry of Education, Culture, Sports, Science and Technology, instead of asking a total number of educational years. This questioning was included in the administration time. Also, our study participants were research participants, and they might have been more talkative than patients and their families attending a clinic for assessment.

The relatively low accuracy of the Q*mci*-J screen for cognitive impairment is in contrast with other studies showing good to excellent accuracy for the Q*mci* screen [15,16,17,26,27]. Yet despite the fact that the NEIGE cohort study does not collect information sufficient to diagnose patients with cognitive disorders according to standardized criteria (e.g., DSM-5) and the inherent issues of using the sMMSE-J to categorize impairments, the Q*mci-*J screen still showed moderate diagnostic agreement. The authors then adjusted the sMMSE scores for age in an attempt to improve the diagnostic classification, and found that the accuracy of the Q*mci*-J screen further improved.

In this study because the accuracy of the Q*mci*-J screen was dependent on the classification of cognitive impairment based on sMMSE-J scores, it was not possible to compare the diagnostic accuracy of the two instruments. This is another limitation of the available data and the subsequent analysis in the paper, and is important because the sMMSE-J was not designed to detect MCI, unlike the Q*mci*-J screen [27,28,29] In order to evaluate the clinical diagnostic accuracy of the Q*mci*-J screen in comparison with the MMSE, future studies should compare the Q*mci*-J and predictions by MMSE-J of subjects with cognitive impairment diagnosis based on clinical and neuropsychological examination.

The optimal Q*mci*-J screen cut-off for predicting those with suspected cognitive impairment identified by MMSE-J was 60/61 (<61/100). This is very similar to the proposed cut-off for the original English version with respect to predicting cognitive impairment (MCI and dementia) identified by virtue of clinical and neuropsychological examination [26,27]. A similar cut-off, 62 (<62/100) provided a sensitivity of 83% and specificity of 87% in an Irish sample [26], and the identical cut-off, 61 (<61/100) produced a sensitivity of 90% and specificity of 87% in a Canadian sample [27]]. The optimal Q*mci*-J cut-off for predicting suspected dementia identified by sMMSE-J was 56 (<56/100). This is higher than the proposed cut-off for the original English version, <54/100, though it is again similar [27]. It must be highlighted that our data was normative data, and only 3 subjects (0.57%) were currently receiving medical care for dementia. Higher cut-off points may be efficient enough to discriminate suspected cognitive impairment from normal cognition in community-dwelling older adults. Further study is now required to examine the diagnostic accuracy against a more clinically appropriate classification of cognitive impairment using internationally recognized criteria, and against MCI-specific instruments including the Japanese version of the Montreal Cognitive Assessment (MoCA) [28] in different healthcare settings.

This paper is also one of the first, after the recent publication of the Italian version of the Q*mci* screen [20], to provide normative data for the Q*mci* screen in likely normal subjects, and this study applied a similar approach to identify likely normal participants as the Italian paper [20]. Despite the challenges of defining normative data based on sMMSE scores, there was no marked difference in median Q*mci*-J screen scores, after maximizing the likelihood of selecting a truly cognitively normal sample by excluding those with symptoms (both subjective and informant reported), ADL impairment or conditions known to be associated with MCI and dementia, such as stroke and Parkinson’s disease. This suggests that a high proportion of those presumed to have normal cognition based on the sMMSE-J scores likely did have normal cognition, increasing the chance that the results of ROC curve analysis are robust, though this is a limitation of the study. Another limitation is the relatively low PPV, reflecting the cut-off selected [29] and that the sample is community-based and largely asymptomatic with a low prevalence of cognitive impairment [30]. In this setting, the Q*mci*-J would be ideally used as a quick ‘pre-screen’ as part of a two-step screening and assessment approach to diagnosing cognitive impairment. The normative data shown here for an older Japanese population were remarkably consistent with the Italian sample [20]. For example in Italy, those aged 60–69 with 13+ years in education had a mean score of 70/100 on the Q*mci*-I screen [20], the same median score for Japanese participants aged 65–69. For Italians between 70–80 years with 13+ years education, 63.86 was the mean score [20] compared to a median score of 63.5 for those in Japan with the same time in education aged 75–79 years.

## 5. Conclusions

In conclusion, this study presents the concurrent validity of the Japanese version of the Q*mci* screen, showing that it correlates moderately, but has a shorter administration time than the most widely used CSI in Japan, the sMMSE-J. Although further research is required to demonstrate diagnostic accuracy, the Q*mci*-J screen is a short CSI that could be useful from a public health perspective to rapidly determine the likelihood of having cognitive impairment, and whether to proceed with further evaluation and with subsequent diagnostic tests or procedures in community-dwelling older adults at population level, either in acute hospital care or in busy primary care settings.

## Figures and Tables

**Figure 1 ijerph-16-00917-f001:**
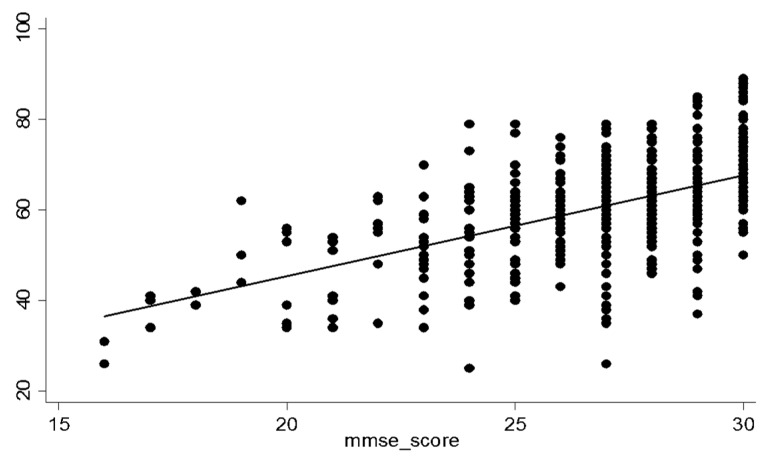
Scatter plot depicting correlations between the standardized MMSE-J and the Q*mci*-J screen.

**Figure 2 ijerph-16-00917-f002:**
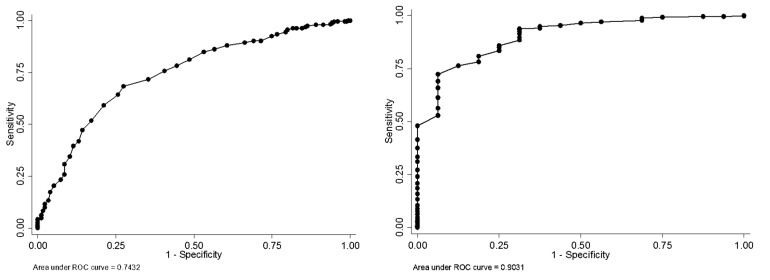
Receiver Operating Characteristic curve for the Quick Mild Cognitive Impairment (Q*mci*-J) screen to detect those with suspected Mild cognitive impairment (MCI) or dementia (**left**) and those with suspected dementia (**right**).

**Table 1 ijerph-16-00917-t001:** Characteristics of all participants included in the analysis along with a breakdown of those categorized with suspected cognitive impairment using established cut-off scores on the standardized Mini-mental State Examination.

	Total (n = 526)	Suspected Dementia (n = 16)	Suspected MCI (n = 214)	Normal Cognition (n = 351)	*p*-Value *
	% or Mean ± SD	% or Mean ± SD	% or Mean ± SD	% or Mean ± SD	
**Sex**					0.051
Male	47.3	56.3	54.7	43.6	
Female	52.7	43.8	45.3	56.4	
**Age (years)**	73.5 ± 5.6	78.0 ± 5.3	75.2 ± 5.6	72.5 ± 5.3	<0.001
**Education**					<0.001
Elementary school	1.3	6.3	3.8	0.0	
Junior high school	40.3	81.3	52.2	33.1	
High school	44.7	12.5	35.2	50.4	
Some college/university/graduate school	13.7	0.0	8.8	16.5	
**Visual Impairment**	1.0	0.0	0.6	1.1	0.89
**Hearing Impairment**	2.7	18.8	2.5	2.0	<0.001

MCI = mild cognitive impairment; SD = standard deviation.

**Table 2 ijerph-16-00917-t002:** Sensitivity (Sen), Specificity (Spe), Positive Predictive Value (PPV), Negative Predictive Value (NPV, with 95% confidence intervals, for different Quick Mild Cognitive Impairment (Q*mci*-J) screen cut-off scores for suspected cognitive impairment (suspected MCI and dementia), without adjustment for age or education, compared with normal controls.

Cut-off	Youden’s Index (J)	Sen (95%CI)			Spe (95%CI)			PPV (95%CI)			NPV (95%CI)		
**54/55**	0.29	0.43	0.36	0.51	0.86	0.82	0.90	0.61	0.52	0.70	0.75	0.71	0.80
**55/56**	0.32	0.47	0.39	0.55	0.85	0.81	0.89	0.61	0.52	0.69	0.76	0.72	0.80
**56/57**	0.32	0.51	0.44	0.59	0.81	0.77	0.85	0.58	0.50	0.66	0.77	0.72	0.81
**57/58**	0.33	0.55	0.48	0.63	0.78	0.74	0.83	0.56	0.48	0.64	0.78	0.73	0.82
**58/59**	0.35	0.59	0.52	0.67	0.76	0.71	0.80	0.55	0.48	0.62	0.79	0.74	0.83
**59/60**	0.37	0.65	0.57	0.72	0.72	0.67	0.76	0.53	0.46	0.60	0.80	0.75	0.85
**60/61**	**0.41**	0.73	0.65	0.79	0.68	0.63	0.73	0.53	0.47	0.60	0.83	0.79	0.87
**61/62**	0.38	0.74	0.67	0.81	0.64	0.59	0.69	0.51	0.45	0.57	0.83	0.78	0.88
**62/63**	0.38	0.79	0.72	0.85	0.59	0.54	0.64	0.49	0.43	0.55	0.85	0.80	0.89
**63/64**	0.35	0.83	0.76	0.88	0.52	0.47	0.57	0.46	0.41	0.52	0.86	0.80	0.90
**64/65**	0.33	0.86	0.80	0.91	0.47	0.42	0.53	0.45	0.39	0.50	0.87	0.81	0.91
**65/66**	0.29	0.87	0.81	0.92	0.42	0.37	0.47	0.43	0.38	0.48	0.87	0.80	0.91

Note: optimal value highlighted in bold.

**Table 3 ijerph-16-00917-t003:** Sensitivity (Sen), Specificity (Spe), Positive Predictive Value (PPV), Negative Predictive Value (NPV, with 95% confidence intervals (CI), for different Quick Mild Cognitive Impairment (Q*mci*-J) screen cut-off scores for suspected cognitive impairment (suspected dementia), without adjustment for age or education, compared with normal controls.

Cut-off	Youden’s Index (J)	Sen (95%CI)			Spe (95%CI)			PPV (95%CI)			NPV (95%CI)		
**40/41**	0.47	0.50	0.25	0.75	0.97	0.95	0.98	0.31	0.14	0.52	0.98	0.97	0.99
**41/42**	0.51	0.56	0.30	0.80	0.95	0.93	0.97	0.27	0.13	0.46	0.99	0.97	0.99
**42/43**	0.58	0.63	0.35	0.85	0.95	0.93	0.97	0.28	0.14	0.45	0.99	0.97	1.00
**43/44**	0.57	0.63	0.35	0.85	0.94	0.92	0.96	0.25	0.13	0.41	0.99	0.97	1.00
**44/45**	0.63	0.69	0.41	0.89	0.94	0.91	0.96	0.26	0.14	0.41	0.99	0.98	1.00
**45/46**	0.62	0.69	0.41	0.89	0.93	0.91	0.95	0.24	0.13	0.40	0.99	0.98	1.00
**46/47**	0.61	0.69	0.41	0.89	0.92	0.90	0.95	0.22	0.12	0.36	0.99	0.98	1.00
**47/48**	0.60	0.69	0.41	0.89	0.91	0.89	0.94	0.20	0.10	0.33	0.99	0.98	1.00
**48/49**	0.59	0.69	0.41	0.89	0.90	0.87	0.92	0.17	0.09	0.29	0.99	0.98	1.00
**49/50**	0.57	0.69	0.41	0.89	0.88	0.85	0.91	0.16	0.08	0.26	0.99	0.98	1.00
**50/51**	0.61	0.75	0.48	0.93	0.86	0.83	0.89	0.14	0.08	0.24	0.99	0.98	1.00
**51/52**	0.60	0.75	0.48	0.93	0.85	0.82	0.88	0.14	0.07	0.23	0.99	0.98	1.00
**52/53**	0.59	0.75	0.48	0.93	0.84	0.80	0.87	0.13	0.07	0.21	0.99	0.98	1.00
**53/54**	0.62	0.81	0.54	0.96	0.81	0.77	0.84	0.12	0.06	0.19	0.99	0.98	1.00
**54/55**	0.59	0.81	0.54	0.96	0.78	0.74	0.82	0.11	0.06	0.17	0.99	0.98	1.00
**55/56**	0.64	0.88	0.62	0.98	0.76	0.72	0.80	0.10	0.06	0.17	1.00	0.98	1.00
56/57	**0.66**	0.94	0.70	1.00	0.72	0.68	0.76	0.10	0.05	0.15	1.00	0.99	1.00
**57/58**	0.63	0.94	0.70	1.00	0.69	0.65	0.73	0.09	0.05	0.14	1.00	0.98	1.00
**58/59**	0.60	0.94	0.70	1.00	0.66	0.62	0.70	0.08	0.05	0.13	1.00	0.98	1.00
**59/60**	0.55	0.94	0.70	1.00	0.61	0.57	0.66	0.07	0.04	0.11	1.00	0.98	1.00
**60/61**	0.50	0.94	0.70	1.00	0.56	0.52	0.61	0.06	0.04	0.10	1.00	0.98	1.00
**61/62**	0.47	0.94	0.70	1.00	0.53	0.49	0.57	0.06	0.03	0.10	1.00	0.98	1.00

Note: optimal value highlighted in bold.

**Table 4 ijerph-16-00917-t004:** Total Quick Mild Cognitive Impairment (Q*mci*-J) screen median scores (IQR) by age and education for presumed normative data based on with standardized Mini-Mental State Examination (sMMSE-J) scores ≥27 (n = 352).

Education (years)	Age (years)			
	65–69	70–74	75–79	80–84
6–9	60 (52 to 66)	63 (57 to 78)	62.5 (57 to 66)	61 (55 to 65)
10–12	67 (61 to 73)	66 (61 to 70.5)	63 (58 to 67)	58 (53 to 67)
13+	70 (63 to 76)	69 (64.5 to 73)	66 (60 to 70)	59 (59 to 70)

**Table 5 ijerph-16-00917-t005:** Total Quick Mild Cognitive Impairment (Q*mci*-J) screen median scores (IQR) by age and education based on with sMMSE-J scores ≥27 for those without symptoms of cognitive impairment or suggestive medical conditions or medications (n = 265).

Education (years)	Age (years)			
	65–69	70–74	75–79	80–84
6–9	62 (56 to 66)	63 (56 to 68)	63 (58 to 66)	61.5 (54 to 65)
10–12	67 (62 to 73)	67 (62 to 70.5)	63.5 (58 to 67)	58 (53 to 67)
13+	70 (63 to 75)	65 (63 to 74)	63.5 (61.5 to 65.5)	59 (59 to 70)

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
