# Peer review of "Validity of the Japanese Version of the Quick Mild Cognitive Impairment Screen"

_ijerph, 2019, doi:10.3390/ijerph16060917_

Round 1

Reviewer 1 Report

Authors  did a good job in translating and using a new tool in Japanese population.

But results are not supporting their hypothesis, especially positive predictive value (PPV=53%) is comparatively low among Japanese people compared with other populations.

Almost half of the patients among those who will found positive for disease are actually disease-free. It will unnecessarily increase the financial burden  on health system.

Authors  should provide explanation to rectify this problem and merits of this  study in the light of this extra burden on health care system.

Author Response

Response to comments from the reviewer#1

Point 1                                                                                            Authors did a good job in translating and using a new tool in Japanese population. But results are not supporting their hypothesis, especially positive predictive value (PPV=53%) is comparatively low among Japanese people compared with other populations. Almost half of the patients among those who will found positive for disease are actually disease-free. It will unnecessarily increase the financial burden on health system. Authors should provide explanation to rectify this problem and merits of this study in the light of this extra burden on health care system.

Response to the point

Thank you very much for your review of our manuscript. The authors agree that the low PPV is a relative limitation of the use of the Qmci-J as a screening test and have acknowledged this as a limitation to the discussion section. However, the low PPV also reflects the sample under assessment and the cut-off score taken to maximise sensitivity and specificity using Youden’s index (O’Caoimh et al., 2017). Unlike in other studies using short CSI’s that have been conducted in memory clinics/other clinically-based settings, this study is examining its use in a largely asymptomatic sample (i.e. low prevalence of cognitive impairment) (see Maxim et al., 2014). The low PPV is a reflection of this as much as it is due to the psychometric properties of the Qmci-J. In this sample/setting, the Qmci-J would be ideally used as a quick ‘pre-screen’ as part of a two-step screening and assessment approach to diagnosing cognitive impairment in the community. In this sense the NPV would be of more importance.

To clarify, we have emphasized in the manuscript that the sMMSE and the Qmci-J are both screening tools to determine the likelihood of having the disease, rather than diagnostic tools and the present study aims to examine the Qmci-J’s usefulness among community-dwelling older adults without symptoms.

(line 55-56) A screening test aims at determining the likelihood of having the disease among people who as yet have no symptoms. The most commonly used brief cognitive screening instrument (CSI) is …

(line 236-238) The present study is the first to develop and evaluate a Japanese version of the Qmci-J screen to aid in the screening community-dwelling older adults and determining their likelihood of having cognitive impairment.

(line 300-304) Another limitation is the relatively low PPV, reflecting the cut-off selected [29] and that the sample is community-based and largely asymptomatic with a low prevalence of cognitive impairment [30]. In this setting, the Qmci-J would be ideally used as a quick ‘pre-screen’ as part of a two-step screening and assessment approach to diagnosing cognitive impairment.

(line 313-317) Although further research is required to demonstrate diagnostic accuracy, the Qmci-J screen is a short CSI that could be useful from a public health perspective to rapidly determine likelihood of having cognitive impairment in community-dwelling older adults at population level and whether to proceed with further evaluation and subsequent diagnostic tests or procedures, in acute hospital care or in busy primary care settings.

The following references has been added:

[29] O’Caoimh R, Gao Y, Svendovski A, Gallagher P, Eustace J, Molloy DW. Comparing approaches to optimize cut-off scores for short cognitive screening instruments in mild cognitive impairment and dementia. Journal of Alzheimer's Disease 2017, 57(1):123-33.

[30] Maxim LD, Niebo R, Utell MJ. Screening tests: a review with examplesInhalation toxicology 2014, 26(13): 811-828.

Reviewer 2 Report

The limitations of the paper are presented and I suggest in the next study the neurocognitive diagnosis need to be considered.

Author Response

Response to comments from the reviewer#2

Point 1

The limitations of the paper are presented and I suggest in the next study the neurocognitive diagnosis need to be considered.

Response to point 1:

Following your comments, we proposed that the future studies need to compare the Qmci-J’s prediction with neurocognitive diagnosis of cognitive impairment.

(line 264-266) In order to evaluate the clinical diagnostic accuracy of Qmci-J screen in comparison with the MMSE, future studies should compare the Qmci-J and MMSE-J’s prediction of subjects with cognitive impairment diagnosis based on clinical and neuropsychological examinations.

Reviewer 3 Report

Well written manuscript. There are small spelling and syntax sentences (see lines 23-27; and 51).

Although the reader believes tables 2 and 3 are important data on the study, the reader is not clear of the importance of indexes for specificity and significance  to be included as part of the manuscript for each of the cut off scores.  It may not appear to be reader friendly to the larger audience.

Author Response

Response to Comments from the Reviewer #3

Point 1

Well written manuscript. There are small spelling and syntax sentences (see lines 23-27; and 51).

Point 1                                                                                                                                             Well written manuscript. There are small spelling and syntax sentences (see lines 23-27; and 51).

Response to Point 1:

Thank you very much for taking your time to review our manuscript.

Following your comments, we checked the spelling and syntax of the sentences on line 23-27 and 51, and corrected as below:

(line 23-27) Impairment (Qmci-J) screen, a performance-based easy-to-use, valid and reliable short cognitive screening instrument, and examined validity. Community-dwelling adults aged 65-84 in Niigata prefecture, Japan were concurrently administered the Qmci-J and the Japanese version of the standardized Mini-Mental State Examination (sMMSE-J). Mild Cognitive Impairment (MCI) and dementia were categorized using established and age-adjusted sMMSE-J cut-offs. The sample

(line 54* previously 51) languages including Japanese.[7, 8] The sMMSE-J is a 30-point test that assesses a board range of

Point 2                                                                                                                                        Although the reader believes tables 2 and 3 are important data on the study, the reader is not clear of the importance of indexes for specificity and significance to be included as part of the manuscript for each of the cut off scores. It may not appear to be reader friendly to the larger audience.

Response to Point 2:

According to the reviewer’s advice, we selected the most important parts of the indexes (optimal Youden’s Index and close values) rather than showing the whole. Table 2 and Table 3 are now significantly reduced in size.
